# PO3AD: PREDICTING POINT OFFSETS TOWARD BETTER 3D POINT CLOUD ANOMALY DETECTION

## ABSTRACT

Point cloud anomaly detection, particularly under the anomaly-free setting, poses a significant challenge as it requires the precise capture of 3D normal data features to accurately identify deviations indicative of anomalies. Current efforts focus on devising reconstruction tasks, such as acquiring normal data representations by restoring normal samples from altered, pseudo-anomalous counterparts. Nonetheless, such methods tend to dilute the model's focus, as they require attention to both normal and pseudo-anomalous data points, thereby hampering the efficacy of the learning process. Moreover, the inherently disordered and sparse nature of 3D point cloud data significantly complicates the task. In response to those predicaments, we introduce an innovative approach that involves learning *point offsets* for the first time, with a concentrated emphasis on more informative pseudo-abnormal points, thus fostering more effective distillation of normal data representations. We have crafted an augmentation technique that is steered by *normal vectors*, facilitating the creation of credible pseudo anomalies that enhance the efficiency of the training process. Our comprehensive experimental evaluation on the Anomaly-ShapeNet and Real3D-AD datasets evidences that our proposed method outperforms existing state-of-the-art approaches, achieving an average enhancement of 9.0% and 1.4% in the AUC-ROC detection metric across these datasets, respectively.

## 1 INTRODUCTION

Point cloud anomaly detection aims to identify defective samples and locate abnormal regions that deviate from expected data patterns (Roth et al., 2022; Zhou et al., 2024). Owing to the high cost of collecting and labeling anomaly samples, this task is usually implemented in an anomaly-free setting, i.e., only normal samples are available during training. The critical challenge within this framework is to effectively capture the distinctive features that are characteristic of 3D normal data, enabling the system to recognize and classify instances that deviate from these normal patterns as anomalies. Nonetheless, the inherently disordered and sparse nature of 3D point cloud data significantly complicates the process of acquiring such discriminative knowledge.

As one reasonable way to tackle this task, anomaly detection in 3D point clouds often involves designing reconstruction tasks to capture normal representations,as illustrated in Fig. 1(a). Anomalies are detected by comparing inputs to their reconstruction outputs. For instance, IMRNet (Li et al., 2024) randomly masks training normal samples and trains a reconstruction task to restore complete point clouds. However, this approach may fail to detect anomalies in unmasked regions. To address this limitation, R3D-AD (Zhou et al., 2024) proposes reconstructing normal samples from their pseudo-abnormal variants. A test sample with high differences between its input and output is considered an anomaly. Despite its efficacy, reconstructing each point's coordinates in 3D space causes the model to assign equal loss weight to both normal and pseudo-abnormal points, which may hinder learning normal representations. Empirical evidence in Fig. 1(c) shows that the performance degrades as the normal point loss weight increases from 0.1 to 1.0 (the loss weight of pseudo-abnormal points is fixed at 1.0). Extraction of normal patterns relies on learning to restore normal regions from pseudo-abnormal ones, but equal loss weights impair the network to focus on this process, thus limiting the detection performance.

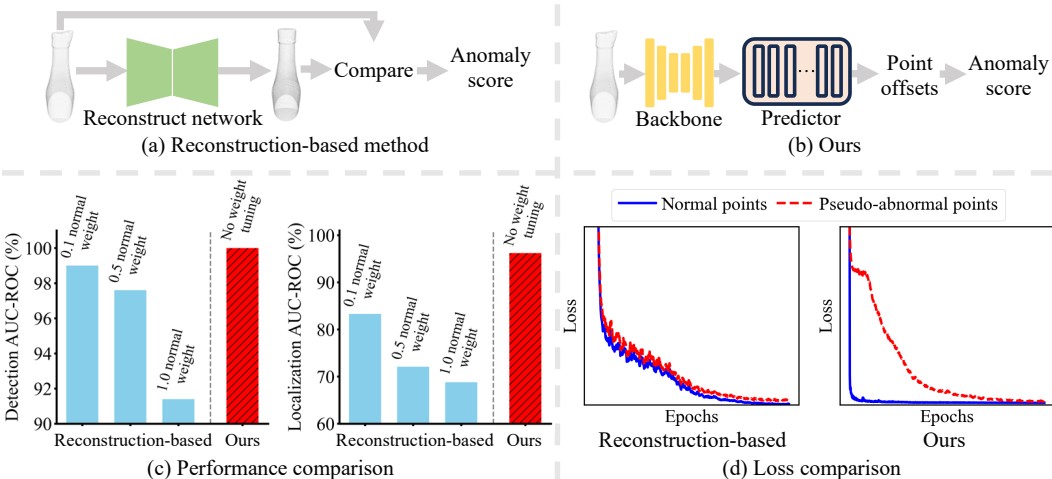

Figure 1: Comparison of reconstruction-based method and our method in terms of structure, performance, and efficiency. (a) Restores normal samples from pseudo-abnormal variants; anomaly scores from input-output comparison. (b) Predicts point offsets of pseudo anomalies; anomaly scores from predicted offsets during testing. (c) Detection and localization performance of the reconstruction-based method on the ashtray0 category with various normal point loss weights; pseudo-abnormal points consistently weighted at 1.0 (implemented with our network due to the absence of official code). (d) Our method quickly converges on normal points, enabling focus on anomalies in later training (loss values are normalized to range 0-1 using min-max method).

In this paper, we propose to predict point offsets for pseudo anomalies (as illustrated in Fig. 1(b)) to allow the model to concentrate on pseudo-abnormal regions, ensuring the effective distillation of normal representations. Specifically, point offsets are essentially vectors characterized by two attributes: magnitude and direction. The offsets of abnormal points in pseudo anomalies are defined by these attributes, representing their displacement distance and direction relative to their corresponding points in original normal ones. In contrast, the offsets of normal points in pseudo anomalies can be predominantly governed by their displacement distance, as they remain unchanged relative to their corresponding points in original normal ones, making the direction less relevant and the magnitude zero. Therefore, learning the task of point offset prediction allows the model to estimate normal points' offset magnitude only, while requiring it to predict both offset magnitude and direction for pseudo-abnormal points. This is significantly different from the current mainstream reconstruction-based methods that need to precisely restore the coordinates of each point, thus leading the model to concentrate unnecessarily on both normal and pseudo-abnormal points simultaneously. Empirical evidence is presented in Fig. 1(d). In the right part, losses converge faster on normal points than on pseudo-abnormal points, enabling the model to focus on pseudo-abnormal points in late training. However, the losses of normal points follow almost the same trend as those of pseudo-abnormal points in the reconstruction-based method, i.e., the model equally concentrates on both two kinds of points. Additionally, the predicted offsets of test samples can directly assess their abnormality levels during inference, while reconstruction-based methods need to design handcrafted metrics to produce anomaly scores.

Drawing inspiration from the aforementioned observation, we propose a novel framework named PO3AD, which efficiently predicts point offsets and adequately captures normal representations. For practical implementation, in order to enable the model to learn the knowledge of predicting offsets, we further propose an anomaly simulation method named Norm-AS, which is guided by *normal vectors* [1]. Norm-AS is performed by moving points of a random region in normal data along or against the *normal vectors* to produce pseudo anomalies. In contrast, the previous augmentation method (Zhou et al., 2024) ignores point movement direction, resulting in points potentially moving in any direction in 3D space. This may cause pseudo-abnormal regions to overlap with normal

---

[1]In this paper, '*normal vectors*' exclusively refers to the vectors perpendicular to the surface in point cloud geometry, while 'normal' denotes non-abnormal. To avoid confusion, we italicized *normal vectors*.

regions (as shown in Fig. 3(c)), which consequently confuses the model, leading to less effective learning. Our Norm-AS leverages *normal vectors* to control point movement direction, enabling the creation of credible pseudo anomalies that resemble real ones (as shown in Fig. 3(d)), thus increasing learning efficiency. The offsets of points in pseudo anomaly samples relative to their original normal counterparts serve as training labels. During testing, the predicted offsets are used to recognize anomalies.

Our contributions can be summarized as follows:

- We propose a novel paradigm named PO3AD to predict point offsets, allowing the model to concentrate on pseudo-abnormal regions and ensuring the effective learning of normal representations for 3D point cloud anomaly detection.
- We design a point cloud pseudo anomaly generation method guided by *normal vectors*, termed Norm-AS, creating credible pseudo anomalies from normal samples for improving training efficiency.
- Extensive experiments conducted on two benchmark point cloud anomaly detection datasets demonstrate the superiority of our method to state-of-the-art methods, with an average improvement of 9.0% and 1.4% detection AUC-ROC on Anomaly-ShapeNet and Real3D-AD, respectively.

## 2 RELATED WORK

**2D anomaly detection.** Anomaly detection methods on 2D image data under anomaly-free scenarios have been widely studied in recent years. To address the issue that anomalies are unavailable during training, a straightforward approach involves generating pseudo anomalies (Hu et al., 2024; Zavrtanik et al., 2021a; Li et al., 2021; Schlüter et al., 2022; Liu et al., 2023b; Zhang et al., 2024), allowing models to learn discriminative knowledge for identifying anomalies. An alternative way to tackle this task relies on constructing a memory bank storing normal features produced by pretrained encoders (Bae et al., 2023; Kim et al., 2023; Roth et al., 2022; Xie et al., 2023). Such methods detect anomalies by contrasting features of test data with those of normal training samples. Flow-based methods (Rudolph et al., 2021; Gudovskiy et al., 2022) leverage normalizing flows for estimation of the feature distribution to detect anomalies. Reconstruction-based methods (Huang et al., 2022; Pirnay & Chai, 2022; Yan et al., 2021; Zavrtanik et al., 2021b) designs reconstruction tasks to capture normal representations; anomalies are detected by comparing inputs to their reconstruction results. In this paper, we focus on 3D point cloud anomaly detection. This task is particularly challenging due to the disordered and sparse characteristics of point cloud data.

**3D anomaly detection.** Although significant progress has been made in 2D anomaly detection, research into anomaly detection for 3D data is still relatively limited. Due to the absence of point cloud anomaly detection datasets, early studies are conducted on RGB-D datasets, such as the MVTec AD-3D dataset (Bergmann et al., 2022). AST (Rudolph et al., 2023) enhances the detection capability by leveraging depth information to suppress background. 3D-ST (Bergmann & Sattlegger, 2023) proposes a teacher-student framework to capture representations of normal samples during training, and anomalies are detected by assessing regression errors between teacher and student networks. BTF (Horwitz & Hoshen, 2023) proposes to utilize handcrafted 3D descriptors combined with K-Nearest Neighbors (KNN) to tackle the task of 3D anomaly detection. M3DM (Wang et al., 2023) designs a multimodal hybrid fusion paradigm that merges point and image features to strengthen the detection performance. CPMF (Cao et al., 2024) fuses 2D and 3D features by projecting point cloud data into multi-view images to construct a memory bank. With the proposal of two point cloud anomaly detection datasets: Real3D-AD (Liu et al., 2023a) and Anomaly-ShapeNet (Li et al., 2024), recent efforts focus on anomaly detection for point cloud data. Reg3D-AD combines the classical 2D method PatchCore (Roth et al., 2022) with RANSAC algorithm (Bolles & Fischler, 1981) to develop a memory bank-based framework for point cloud anomaly detection. Group3AD (Zhu et al., 2024) groups points into multiple clusters and designs a group-level contrastive loss to capture inter-cluster dispersion and intracluster compactness features, which are subsequently stored in a memory bank. Although memory bank-based methods have shown effectiveness, they suffer the prohibitive computational and storage. IMRNet (Li et al., 2024) adopts the idea of 2D reconstruction-based methods, randomly masking training point clouds and restoring them by training a PointMAE (Pang et al., 2022). While R3D-AD (Zhou et al., 2024) creates pseudo anomalies from normal samples

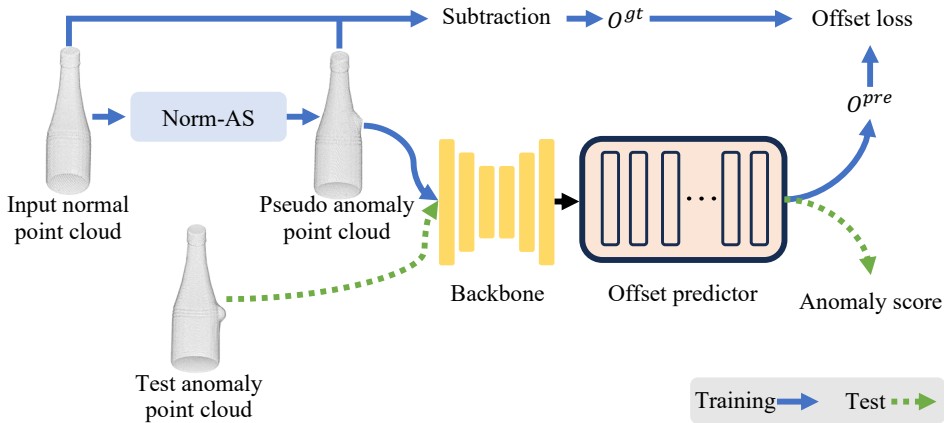

Figure 2: Illustration of our framework. Norm-AS generates pseudo anomalies from training normal samples. The backbone extracts features from pseudo anomalies, and the offset predictor estimates offsets for each point of input. The network trains under an offset loss constraint. During inference, the predicted offset distances serve as anomaly scores for test instances.

and reconstructs them via a denoising Diffusion model (Ho et al., 2020), anomalies are detected by evaluating the differences between inputs and their outputs. Unlike previous methods, we make a first attempt and propose to predict point offsets to capture effective normal representations.

## 3 METHODOLOGY

**Problem statement.** Point cloud anomaly detection involves a training set $\mathcal{D}_{train}^e = \{P_q \in \mathbb{R}^{N \times 3}\}_{q=1}^M$, which consists of $M$ normal samples with $N$ points, belonging to a specific category $e$. A test set, $\mathcal{D}_{test}^e = \{P_q \in \mathbb{R}^{N \times 3}, t_q \in \mathcal{T}\}_{q=1}^K$, consists of samples $P_q$ with labels $t_q$, where $\mathcal{T} = \{0, 1\}$ (0 denotes a normal and 1 denotes an anomaly). The objective is to train a deep anomaly detection model on $\mathcal{D}_{train}^e$ to build a scoring function $\phi: \mathbb{R}^{N \times 3} \to \mathbb{R}$ that quantitatively evaluate the abnormality levels of new point cloud instances.

**Overview.** The overview of our framework is presented in Fig. 2. Given one sample for illustrating our procedure, a pseudo anomaly point cloud is generated from it by our Norm-AS. The subtraction of the input normal sample from the pseudo-abnormal one is used as the training label. Then, the pseudo anomaly is fed into a backbone to extract its features. An offset prediction module then takes these features as input to produce the prediction results. Afterward, the model parameters are optimized by an offset loss. During testing, the predicted offsets are applied to test data to evaluate their abnormal levels.

### 3.1 OFFSET PREDICTION LEARNING

To capture normal representation for anomaly detection, we propose to predict point offsets. Practically, we construct an offset prediction network and leverage an offset loss to supervise the network in learning the knowledge of estimating points offsets.

#### 3.1.1 OFFSET PREDICTION NETWORK

Our network is composed of two modules: a backbone and an offset predictor. Inspired by exemplary pioneering work (Hu et al., 2021; Zhao et al., 2023; Schult et al., 2023; Delitzas et al., 2024) in 3D domain, we adopt MinkUNet (Choy et al., 2019b;a) as the backbone for our method. Specifically, MinkUNet is a voxel-based sparse convolutional network (Graham, 2015; Gwak et al., 2020) that effectively captures detailed local features from point clouds. This allows the extraction of fine-grained pseudo-abnormal features during training, thus facilitating normal representation learning. Given one point cloud sample $P \in \mathbb{R}^{N \times 3}$, it is voxelized into $V \in \mathbb{R}^{N_V \times 3}$, where $N_V$ stands for the number of voxels. It is noted that $N_V \leq N$ and $N_V$ are inversely correlated with the voxel size.

The MinkUNet $f_U$ maps $V$ to latent voxelized features $G^V \in \mathbb{R}^{N_V \times C} = f_U(V)$, where $C$ denotes the dimension of each voxel's feature. Then, the voxel-to-point index is leveraged to transform $G^V$ to latent point features $G^P \in \mathbb{R}^{N \times C}$, which are utilized to predict point-wise offsets. Our offset predictor $f_O$ is constructed using a Multi-Layer Perceptron (MLP), which takes $G^P$ as input to estimate the offset of each point $O^{pre} \in \mathbb{R}^{N \times 3} = f_O(G^P)$. The offset of each point is composed of three coordinate (xyz) offsets. Each element in $O^{pre}$ refers to the offset of a point along a particular coordinate.

### 3.1.2 OFFSET LOSS

An offset loss is adopted to guide the network in learning the knowledge of predicting point offsets. These point offsets are vectors that describe the displacement distance and direction of each point in pseudo anomalies compared to its corresponding point in normal ones. Accordingly, an L1 loss and a negative cosine loss are employed to supervise the network in predicting point offset distance and direction, respectively, which yields an offset loss:

$$\mathcal{L}_{off} = \mathcal{L}_{dist} + \mathcal{L}_{dir}, \tag{1}$$

$$\mathcal{L}_{dist} = \frac{1}{N} \sum_{i=1}^{N} o_i^{pre} \in O^{pre}, o_i^{gt} \in O^{gt} \left\| o_i^{pre} - o_i^{gt} \right\|, \tag{2}$$

$$\mathcal{L}_{dir} = -\frac{1}{N} \sum_{i=1}^{N} o_i^{pre} \in O^{pre}, o_i^{gt} \in O^{gt} \frac{o_i^{pre}}{\|o_i^{pre}\|_2 + \epsilon} \cdot \frac{o_i^{gt}}{\|o_i^{gt}\|_2 + \epsilon}, \tag{3}$$

where $\mathcal{L}_{dist}$ and $\mathcal{L}_{dir}$ are equally weighted to avoid a possible bias to one loss. Here, $\epsilon$ is set to 1e-8 to prevent division by zero, and $O^{gt} \in \mathbb{R}^{N \times 3} = \hat{P} - P$, where $\hat{P}$ is a pseudo anomaly sample created from $P$ through the Norm-AS. It is worth noting that $L_{dir}$ works for pseudo-abnormal points only since the ground truth offset for each normal point is a zero vector. The significance of $\mathcal{L}_{dist}$ and $\mathcal{L}_{dir}$ in capturing normal representations is demonstrated in Section 4.5.

### 3.2 NORM-AS

To create credible pseudo anomalies to improve training efficiency, we develop a novel anomaly simulation method guided by *normal vectors*. Our proposed Norm-AS is performed by moving the points of a random region along the *normal vectors* or in the opposite direction, generating anomaly types of bulge or concavity. The region is selected by dividing a point cloud into multiple patches and then randomly sampling one of these patches. Given a training normal point cloud sample $P \in \mathbb{R}^{N \times 3}$, it is divided into $J$ patches as $PH = \{ph_b \in \mathbb{R}^{N_h \times 3}\}_{b=1}^{J}$, where $N_h$ is the number of points in each patch and is equal to $N/J$. Specifically, Each patch is determined iteratively by randomly selecting one point and its nearest $N_h - 1$ points from $P^r$. $P^r$ denotes the points in the point cloud $P$ that have not been included in any patches. In light of this, $ph_b$ exhibits various shapes rather than being only circular, enabling the creation of pseudo anomalies with various shapes. A randomly sampled $ph_b$ is then produced as a pseudo-abnormal region by:

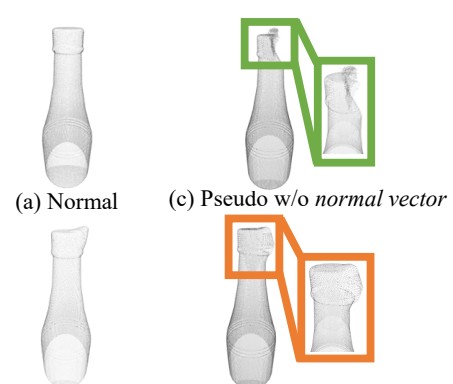

(a) Normal     (c) Pseudo w/o *normal vector*

(b) Real anomaly (d) Pseudo w/ *normal vector* (ours)

Figure 3: Visualization of pseudo samples with and without *normal vectors* on the bottle0 category. Samples generated with *normal vectors* better mimic real anomalies.

$$\hat{ph}_b = ph_b + \alpha \cdot nv_b \cdot (1 - w) \cdot \beta, \tag{4}$$

where $nv_b \in \mathbb{R}^{N_h \times 3}$ is the *normal vectors* of $ph_b$. $\alpha$ is randomly sampled from $\{-1, 1\}$ to control whether the point moves along the $nv_b$ ($\alpha = 1$) or in the opposite direction ($\alpha = -1$). $w$ refers to a matrix with $N_h$ elements, each representing the normalized distance of a point in $ph_b$ from the center

point. By performing $1 - w$, we aim to move the center point the greatest distance, while points farther from the center are moved shorter distances. $\beta$ denotes the movement distance of the center point. It is sampled from a uniform distribution, where the range is empirically set to $[0.06, 0.12]$, to produce pseudo anomalies with various offset distances. A pseudo anomaly sample is produced by replacing the corresponding region in $P$ with $p\hat{h}_b$. The size of the pseudo-abnormal region is determined by $J$, the impact of $J$ for normal representation learning is described in Section 4.6. The Norm-AS enables the creation of pseudo anomalies resembling real ones, as evidenced in Fig. 3(d). As for the pseudo anomaly generated without the guidance of *normal vectors*, as shown in Fig. 3(c), pseudo-abnormal points overlap with normal ones, which may hinder the model from extracting effective features of this region, resulting in training efficiency reduction. More examples of our pseudo anomalies are provided in Fig. 7 of Appendix A. The significance of generating pseudo anomalies guided by *normal vectors* for normal representation learning is validated in Section 4.5.

## 3.3 ANOMALY SCORE FOR INFERENCE

The abnormal level for each point in test data is assessed by its predicted offset. Specifically, the anomaly score of a point is calculated by summing the offset distances along three coordinates (xyz). The point-level anomaly score function $\phi(p_i)$ is defined as:

$$\phi(p_i) = \left| o_{i,x}^{pre} \right| + \left| o_{i,y}^{pre} \right| + \left| o_{i,z}^{pre} \right|, \tag{5}$$

where $p_i \in P$ and $\{o_{i,x}^{pre}, o_{i,y}^{pre}, o_{i,z}^{pre}\} = o_i^{pre} \in O^{pre}$. According to $\phi(p_i)$, the object-level anomaly score function $\phi(P)$ is obtained by:

$$\phi(P) = \frac{1}{N} \sum_{i=1}^{N} \phi(p_i). \tag{6}$$

The anomaly scores for normal samples or points are expected to be as small as possible. The greater the anomaly score, the more likely that a sample or point is an anomaly.

## 4 EXPERIMENTS

### 4.1 EXPERIMENTAL SETTINGS

**Datasets.** Our evaluation encompasses two 3D point cloud anomaly detection datasets: Anomaly-ShapeNet (Li et al., 2024) and Real3D-AD (Liu et al., 2023a). Anomaly-ShapeNet is a synthesis dataset based on ShapeNet (Chang et al., 2015) dataset. It consists of 1,600 samples belonging to 40 categories. The training set of each category contains 4 normal samples, and the test set includes both normal and abnormal samples. Real3D-AD is a high-resolution point cloud dataset based on real objects of 12 categories. Each category contains 4 training normal samples and 100 test instances. There is a large difference between training and test samples in the Real3D-AD dataset where training samples undergo 360° scan, while test samples are scanned on only one side.

**Evaluation metrics.** Experiments are conducted by following previous work (Liu et al., 2023a; Li et al., 2024). Area Under the Receiver-Operating-Characteristic Curve (AUC-ROC) is utilized as our evaluation criterion. It can objectively evaluate detection (object-level) and localization (point-level) performance without making any assumption on the decision threshold.

### 4.2 IMPLEMENTATION DETAILS

The MinkUNet34C (Choy et al., 2019b;a) serves as our backbone for feature extraction. A three-layer MLP with PReLU activation function forms the offset predictor. We set the dimension of latent features $C$ to 32, and the voxel size to 0.03. Our network is trained for 1,000 epochs with a batch size of 32 (the training set is replicated 100 times to obtain 400 samples). The model parameters are optimized by Adam with an initial learning rate of 0.001, which decays with the cosine anneal schedule (Loshchilov & Hutter, 2017). Our method does not involve point cloud downsampling. Training samples are applied with random rotation before normalization. All input point clouds are normalized by aligning their center of gravity with the origin of coordinates and scaling their dimensions to range from -1 to 1. We set the patch number $J$ to 64 for our Norm-AS, which is performed after normalization. The *normal vectors* are obtained from official OBJ files of datasets.

Table 1: Comparison of object-level AUC-ROC results (%) of various methods on the Anomaly-ShapeNet dataset. The best result per category is **bold**, while the second best result is underlined. Micro. refers to the microphone0 category. BTF (Raw) refers to that the point coordinates are adopted into the BTF method. PFFH and PointMAE denote utilizing Fast Point Feature Histograms (Rusu et al., 2009) and ShapeNet (Chang et al., 2015) pre-trained PointMAE (Pang et al., 2022) as the feature extractor, respectively.

| Category | BTF (Raw) (CVPR 23') | BTF (FPFH) | M3DM (CVPR 23') | PatchCore (FPFH) (CVPR 22') | PatchCore (PointMAE) | CPMF (PR 24') | Reg3D-AD (NeurIPS 23') | IMRNet (CVPR 24') | R3D-AD (ECCV 24') | Ours |
|---|---|---|---|---|---|---|---|---|---|---|
| ashtray0 | 57.8 | 42.0 | 57.7 | 58.7 | 59.1 | 35.3 | 59.7 | 67.1 | 83.3 | **100.0** |
| bag0 | 41.0 | 54.6 | 53.7 | 57.1 | 60.1 | 64.3 | 70.6 | 66.0 | 72.0 | **83.3** |
| bottle0 | 59.7 | 34.4 | 57.4 | 60.4 | 51.3 | 52.0 | 48.6 | 55.2 | 73.3 | **90.0** |
| bottle1 | 51.0 | 54.6 | 63.7 | 66.7 | 60.1 | 48.2 | 69.5 | 70.0 | 73.7 | **93.3** |
| bottle3 | 56.8 | 32.2 | 54.1 | 57.2 | 65.0 | 40.5 | 52.5 | 64.0 | 78.1 | **92.6** |
| bowl0 | 56.4 | 50.9 | 63.4 | 50.4 | 52.3 | 78.3 | 67.1 | 68.1 | 81.9 | **92.2** |
| bowl1 | 26.4 | 66.8 | 66.3 | 63.9 | 62.9 | 63.9 | 52.5 | 70.2 | 77.8 | **82.9** |
| bowl2 | 52.5 | 51.0 | 68.4 | 61.5 | 45.8 | 62.5 | 49.0 | 68.5 | 74.1 | **83.3** |
| bowl3 | 38.5 | 49.0 | 61.7 | 53.7 | 57.9 | 65.8 | 34.8 | 59.9 | 76.7 | **88.1** |
| bowl4 | 66.4 | 60.9 | 46.4 | 49.4 | 50.1 | 68.3 | 66.3 | 67.6 | 74.4 | **98.1** |
| bowl5 | 41.7 | 69.9 | 40.9 | 55.8 | 59.3 | 68.5 | 59.3 | 71.0 | 65.6 | **84.9** |
| bucket0 | 61.7 | 40.1 | 30.9 | 46.9 | 59.3 | 48.2 | 61.0 | 58.0 | 68.3 | **85.3** |
| bucket1 | 32.1 | 63.3 | 50.1 | 55.1 | 56.1 | 60.1 | 75.2 | 77.1 | 75.6 | **78.7** |
| cap0 | 66.8 | 61.8 | 55.7 | 58.0 | 58.9 | 60.1 | 69.3 | 73.7 | 82.2 | **87.7** |
| cap3 | 52.7 | 52.2 | 42.3 | 45.3 | 47.6 | 55.1 | 72.5 | 77.5 | 73.0 | **85.9** |
| cap4 | 46.8 | 52.0 | 77.7 | 75.7 | 72.7 | 55.3 | 64.3 | 65.2 | 68.1 | **79.2** |
| cap5 | 37.3 | 58.6 | 63.9 | **79.0** | 53.8 | 69.7 | 46.7 | 65.2 | 67.0 | 67.0 |
| cup0 | 40.3 | 58.6 | 53.9 | 60.0 | 61.0 | 49.7 | 51.0 | 64.3 | 77.6 | **87.1** |
| cup1 | 52.1 | 61.0 | 55.6 | 58.6 | 55.6 | 49.9 | 53.8 | 75.7 | 75.7 | **83.3** |
| eraser0 | 52.5 | 71.9 | 62.7 | 65.7 | 67.7 | 68.9 | 34.3 | 54.8 | 89.0 | **99.5** |
| headset0 | 37.8 | 52.0 | 57.7 | 58.3 | 59.1 | 64.3 | 53.7 | 72.0 | 73.8 | **80.8** |
| headset1 | 51.5 | 49.0 | 61.7 | 63.7 | 62.7 | 45.8 | 61.0 | 67.6 | 79.5 | **92.3** |
| helmet0 | 55.3 | 57.1 | 52.6 | 54.6 | 55.6 | 55.5 | 60.0 | 59.7 | 75.7 | **76.2** |
| helmet1 | 34.9 | 71.9 | 42.7 | 48.4 | 55.2 | 58.9 | 38.1 | 60.0 | 72.0 | **96.1** |
| helmet2 | 60.2 | 54.2 | 62.3 | 42.5 | 44.7 | 46.2 | 61.4 | 64.1 | 63.3 | **86.9** |
| helmet3 | 52.6 | 44.4 | 37.4 | 40.4 | 42.4 | 52.0 | 36.7 | 57.3 | 70.7 | **75.4** |
| jar0 | 42.0 | 42.4 | 44.1 | 47.2 | 48.3 | 61.0 | 59.2 | 78.0 | 83.8 | **86.6** |
| micro. | 56.3 | 67.1 | 35.7 | 38.8 | 48.8 | 50.9 | 41.4 | 75.5 | 76.2 | **77.6** |
| shelf0 | 16.4 | 60.9 | 56.4 | 49.4 | 52.3 | 68.5 | 68.8 | 60.3 | **69.6** | 57.3 |
| tap0 | 52.5 | 56.0 | **75.4** | 75.3 | 45.8 | 35.9 | 67.6 | 67.6 | 73.6 | 74.5 |
| tap1 | 57.3 | 54.6 | 73.9 | 76.6 | 53.8 | 69.7 | 64.1 | 69.6 | **90.0** | 68.1 |
| vase0 | 53.1 | 34.2 | 42.3 | 45.5 | 44.7 | 45.1 | 53.3 | 53.3 | 78.8 | **85.8** |
| vase1 | 54.9 | 21.9 | 42.7 | 42.3 | 55.2 | 34.5 | 70.2 | **75.7** | 72.9 | 74.2 |
| vase2 | 41.0 | 54.6 | 73.7 | 72.1 | 74.1 | 58.2 | 60.5 | 61.4 | 75.2 | **95.2** |
| vase3 | 71.7 | 69.9 | 43.9 | 44.9 | 46.0 | 58.2 | 65.0 | 70.0 | 74.2 | **82.1** |
| vase4 | 42.5 | 51.0 | 47.6 | 50.6 | 51.6 | 51.4 | 50.0 | 52.4 | 63.0 | **67.5** |
| vase5 | 58.5 | 40.9 | 31.7 | 41.7 | 57.9 | 61.8 | 52.0 | 67.6 | 75.7 | **85.2** |
| vase7 | 44.8 | 51.8 | 65.7 | 69.3 | 65.0 | 39.7 | 46.2 | 63.5 | 77.1 | **96.6** |
| vase8 | 42.4 | 66.8 | 66.3 | 66.2 | 66.3 | 52.9 | 62.0 | 63.0 | 72.1 | **73.9** |
| vase9 | 56.4 | 26.8 | 66.3 | 66.0 | 62.9 | 60.9 | 59.4 | 59.4 | 71.8 | **83.0** |
| Average | 49.3 | 52.8 | 55.2 | 56.8 | 56.2 | 55.9 | 57.2 | 66.1 | 74.9 | **83.9** |
| Mean rank | 7.7 | 7.0 | 6.8 | 6.3 | 6.4 | 6.3 | 6.4 | 3.9 | 2.2 | **1.3** |

## 4.3 BASELINE METHODS

We compare our method with eight outstanding methods: BTF (Horwitz & Hoshen, 2023), M3DM (Wang et al., 2023), PatchCore (Roth et al., 2022), CPMF (Cao et al., 2024), Reg3D-AD (Liu et al., 2023a), IMRNet (Li et al., 2024), R3D-AD (Zhou et al., 2024), and Group3AD (Zhu et al., 2024). PatchCore is originally a 2D anomaly detection method and is applied to 3D by replacing feature extractors. The results of BTF, M3DM, PatchCore, and CPMF are implemented by Real3D-AD and IMRNet. The results of other methods are obtained from their papers.

## 4.4 MAIN RESULTS

### 4.4.1 RESULTS ON ANOMALY-SHAPENET

Table 1 and 2 respectively present the detection and localization results of our method alongside the competing methods on the Anomaly-ShapeNet dataset. Evidently, our method achieves the best overall performance on both two tasks, outperforming the second-best method by an average of 9.0% on detection and 23.0% on localization. To prevent a few categories from dominating the averaged results, we also calculate the mean rank (↓) for comparison. Our method obtains the best mean rank on both object-level and point-level AUC-ROC, which is significantly lower than competing methods. At the category level, our method beats competitors in the overwhelming majority

Table 2: Comparsion of point-level AUC-ROC results on the Anomaly-ShapeNet dataset.

| Category | BTF (Raw) (CVPR 23') | BTF (FPFH) | M3DM (CVPR 23') | PatchCore (FPFH) (CVPR 22') | PatchCore (PointMAE) | CPMF (PR 24') | Reg3D-AD (NeurIPS 23') | IMRNet (CVPR 24') | Ours |
|---|---|---|---|---|---|---|---|---|---|
| ashtray0 | 51.2 | 62.4 | 57.7 | 59.7 | 49.5 | 61.5 | 69.8 | 67.1 | **96.2** |
| bag0 | 43.0 | 74.6 | 63.7 | 57.4 | 67.4 | 65.5 | 71.5 | 66.8 | **94.9** |
| bottle0 | 55.1 | 64.1 | 66.3 | 65.4 | 55.3 | 52.1 | 88.6 | 55.6 | **91.2** |
| bottle1 | 49.1 | 54.9 | 63.7 | 68.7 | 60.6 | 57.1 | 69.6 | 70.2 | **84.4** |
| bottle3 | 72.0 | 62.2 | 53.2 | 51.2 | 65.3 | 43.5 | 52.5 | 64.1 | **88.0** |
| bowl0 | 52.4 | 71.0 | 65.8 | 52.4 | 52.7 | 74.5 | 77.5 | 78.1 | **97.8** |
| bowl1 | 46.4 | 76.8 | 66.3 | 53.1 | 52.4 | 48.8 | 61.5 | 70.5 | **91.4** |
| bowl2 | 42.6 | 51.8 | 69.4 | 62.5 | 51.5 | 63.5 | 59.3 | 68.4 | **91.8** |
| bowl3 | 68.5 | 59.0 | 65.7 | 32.7 | 58.1 | 64.1 | 65.4 | 59.9 | **93.5** |
| bowl4 | 56.3 | 67.9 | 62.4 | 72.0 | 50.1 | 68.3 | 80.0 | 57.6 | **96.7** |
| bowl5 | 51.7 | 69.9 | 48.9 | 35.8 | 56.2 | 68.4 | 69.1 | 71.5 | **94.1** |
| bucket0 | 61.7 | 40.1 | 69.8 | 45.9 | 58.6 | 48.6 | 61.9 | 58.5 | **75.5** |
| bucket1 | 68.6 | 63.3 | 69.9 | 57.1 | 57.4 | 60.1 | 75.2 | 77.4 | **89.9** |
| cap0 | 52.4 | 73.0 | 53.1 | 47.2 | 54.4 | 60.1 | 63.2 | 71.5 | **95.7** |
| cap3 | 68.7 | 65.8 | 60.5 | 65.3 | 48.8 | 55.1 | 71.8 | 70.6 | **94.8** |
| cap4 | 46.9 | 52.4 | 71.8 | 59.5 | 72.5 | 55.3 | 81.5 | 75.3 | **94.0** |
| cap5 | 37.3 | 58.6 | 65.5 | 79.5 | 54.5 | 55.1 | 46.7 | 74.2 | **86.4** |
| cup0 | 63.2 | 79.0 | 71.5 | 65.5 | 51.0 | 49.7 | 68.5 | 64.3 | **90.9** |
| cup1 | 56.1 | 61.9 | 55.6 | 59.6 | 85.6 | 50.9 | 69.8 | 68.8 | **93.2** |
| eraser0 | 63.7 | 71.9 | 71.0 | 81.0 | 37.8 | 68.9 | 75.5 | 54.8 | **97.4** |
| headset0 | 57.8 | 62.0 | 58.1 | 58.3 | 57.5 | 69.9 | 58.0 | 70.5 | **82.3** |
| headset1 | 47.5 | 59.1 | 58.5 | 46.4 | 42.3 | 45.8 | 62.6 | 47.6 | **90.7** |
| helmet0 | 50.4 | 57.5 | 59.9 | 54.8 | 58.0 | 55.5 | 60.0 | 59.8 | **87.8** |
| helmet1 | 44.9 | 74.9 | 42.7 | 48.9 | 56.2 | 54.2 | 62.4 | 60.4 | **94.8** |
| helmet2 | 60.5 | 64.3 | 62.3 | 45.5 | 65.1 | 51.5 | 82.5 | 64.4 | **93.2** |
| helmet3 | 70.0 | 72.4 | 65.5 | 73.7 | 61.5 | 52.0 | 62.0 | 66.3 | **84.6** |
| jar0 | 42.3 | 42.7 | 54.1 | 47.8 | 48.7 | 61.1 | 59.9 | 76.5 | **87.1** |
| micro. | 58.3 | 67.5 | 35.8 | 48.8 | **88.6** | 54.5 | 59.9 | 74.2 | 81.0 |
| shelf0 | 46.4 | 61.9 | 55.4 | 61.3 | 54.3 | 78.3 | 68.8 | 60.5 | 66.3 |
| tap0 | 52.7 | 56.8 | 65.4 | 73.3 | 85.8 | 45.8 | 58.9 | 68.1 | 78.3 |
| tap1 | 56.4 | 59.6 | 71.2 | 76.8 | 54.1 | 65.7 | 74.1 | 69.9 | 69.2 |
| vase0 | 61.8 | 64.2 | 60.8 | 65.5 | 67.7 | 45.8 | 54.8 | 53.5 | **95.5** |
| vase1 | 54.9 | 61.9 | 60.2 | 45.3 | 55.1 | 48.6 | 60.2 | 68.5 | **88.2** |
| vase2 | 40.3 | 64.6 | 73.7 | 72.1 | 74.2 | 58.2 | 40.5 | 61.4 | **97.8** |
| vase3 | 60.2 | 69.9 | 65.8 | 43.0 | 46.5 | 58.2 | 51.1 | 40.1 | **88.4** |
| vase4 | 61.3 | 71.0 | 65.5 | 50.5 | 52.3 | 51.4 | 75.5 | 52.4 | **90.2** |
| vase5 | 58.5 | 42.9 | 64.2 | 44.7 | 57.2 | 65.1 | 62.4 | 68.2 | **93.7** |
| vase7 | 57.8 | 54.0 | 51.7 | 69.3 | 65.1 | 50.4 | 88.1 | 59.3 | **98.2** |
| vase8 | 55.0 | 66.2 | 55.1 | 57.5 | 36.4 | 52.9 | 81.1 | 63.5 | **95.0** |
| vase9 | 56.4 | 56.8 | 66.3 | 66.3 | 42.3 | 54.5 | 69.4 | 69.1 | **95.2** |
| Average | 55.0 | 62.8 | 61.6 | 58.0 | 57.7 | 57.3 | 66.8 | 65.0 | **89.8** |
| Mean rank | 6.9 | 4.8 | 5.1 | 5.9 | 6.2 | 6.5 | 3.8 | 4.2 | **1.2** |

of categories, while exhibiting competitive performance in the remaining categories. Additionally, our method attains considerable performance gains compared to the best contestant on various categories, such as bag0 and bowl4. Generally, these comparison results validate the superiority of our method. We also provide object-level AUC-PR results in Table 5 of Appendix B.

### 4.4.2 RESULTS ON REAL3D-AD

Table 3 depicts the comparison of object-level AUC-ROC results on the Real3D-AD dataset. According to the mean rank, our method secures the first place by a narrow margin, with an average AUC-ROC improvement of 1.4% over the second-best method. At the category level, our method achieves the best or the second-best results in 6 categories and exhibits commendable performance in the rest. It is noted that there is a huge gap between training data and test data of the Real3D-AD dataset, i.e., training samples are scanned 360°, but test point clouds are scanned only on one side. The memory bank-based methods (Reg3D-AD, Group3AD) have an advantage when dealing with such situations, as they leverage the technique of template registration to detect anomalies. Despite this, our method still surpasses them on both average performance and mean rank. Compared to reconstruction-based methods, our method achieves the best results in most categories: 8 compared to R3D-AD and 7 compared to IMRNet. Overall, these comparison results evidences the effectiveness of our method.

Table 3: Object-level AUC-ROC results of our method and competitors on the Real3D-AD dataset.

| Category | BTF (Raw) (CVPR 23') | BTF (FPFH) | M3DM (CVPR 23') | PatchCore (FPFH) (CVPR 22') | PatchCore (PointMAE) | CPMF (PR 24') | Reg3D-AD (NeurIPS 23') | IMRNet (CVPR 24') | R3D-AD (ECCV 24') | Group3AD (MM 24') | Ours |
|---|---|---|---|---|---|---|---|---|---|---|---|
| Airplane | 73.0 | 52.0 | 43.4 | **88.2** | 72.6 | 70.1 | 71.6 | 76.2 | 77.2 | 74.4 | 80.4 |
| Car | 64.7 | 56.0 | 54.1 | 59.0 | 49.8 | 55.1 | 69.7 | 71.1 | 69.3 | **72.8** | 65.4 |
| Candy | 53.9 | 63.0 | 55.2 | 54.1 | 66.3 | 55.2 | 68.5 | 75.5 | 71.3 | **84.7** | 78.5 |
| Chicken | 78.9 | 43.2 | 68.3 | 83.7 | 82.7 | 50.4 | **85.2** | 78.0 | 71.4 | 78.6 | 68.6 |
| Diamond | 70.7 | 54.5 | 60.2 | 57.4 | 78.3 | 52.3 | 90.0 | 90.5 | 68.5 | **93.2** | 80.1 |
| Duck | 69.1 | 78.4 | 43.3 | 54.6 | 48.9 | 58.2 | 58.4 | 51.7 | **90.9** | 67.9 | 82.0 |
| Fish | 60.2 | 54.9 | 54.0 | 67.5 | 63.0 | 55.8 | 91.5 | 88.0 | 69.2 | **97.6** | 85.9 |
| Gemstone | 68.6 | 64.8 | 64.4 | 37.0 | 37.4 | 58.9 | 41.7 | 67.4 | 66.5 | 53.9 | **69.3** |
| Seahorse | 59.6 | 77.9 | 49.5 | 50.5 | 53.9 | 72.9 | 76.2 | 60.4 | 72.0 | **84.1** | 75.6 |
| Shell | 39.6 | 75.4 | 69.4 | 58.9 | 50.1 | 65.3 | 58.3 | 66.5 | **84.0** | 58.5 | 80.0 |
| Starfish | 53.0 | 57.5 | 55.1 | 44.1 | 51.9 | 70.0 | 50.6 | 67.4 | 70.1 | 56.2 | **75.8** |
| Toffees | 70.3 | 46.2 | 45.0 | 56.5 | 58.5 | 39.0 | **82.7** | 77.4 | 70.3 | 79.6 | 77.1 |
| Average | 63.5 | 60.3 | 55.2 | 59.3 | 59.4 | 58.6 | 70.4 | 72.5 | 73.4 | 75.1 | **76.5** |
| Men rank | 6.5 | 6.9 | 8.8 | 7.5 | 7.8 | 7.9 | 5.0 | 4.2 | 4.0 | 3.6 | **3.2** |

Table 4: Ablation study of our method and its variants.

| Method | Variant 1 | Variant 2 | Variant 3 | Ours |
|---|---|---|---|---|
| $\mathcal{L}_{dist}$ | ✓ | - | ✓ | ✓ |
| $\mathcal{L}_{dir}$ | - | ✓ | ✓ | ✓ |
| *Normal vector* | ✓ | ✓ | - | ✓ |
| Object-level AUC-ROC | 50.3 | 67.5 | 81.1 | **84.2** |
| Point-level AUC-ROC | 50.4 | 74.9 | 78.4 | **87.8** |

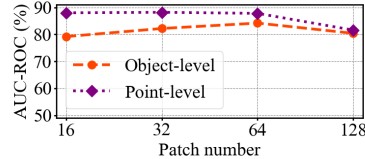

Figure 4: Detection and localization performance *vs.* patch numbers.

## 4.5 ABLATION STUDY

We select fifteen categories ending in 0 of the Anomaly-ShapeNet dataset to conduct the ablation study. The averaged results are reported in Table 4.

**Normal representation learning heavily relies on $\mathcal{L}_{dir}$:** We design "Variant 1", where the model is supervised solely by $\mathcal{L}_{dist}$. The absence of $\mathcal{L}_{dir}$ causes the network to struggle with precisely estimating the offset direction of pseudo-abnormal points. According to the experimental results, the performance of "Variant" is much lower than that of our method, validating the significance of $\mathcal{L}_{dir}$ for capturing effective normal representations.

**$\mathcal{L}_{dist}$ is essential for capturing effective normal representations:** "Variant 2" learns a single objective of predicting point offset direction. Evidently, it is significantly inferior to our method. Without $L_{dist}$, the model fails to learn offset distance for both normal and pseudo-abnormal points. Additionally, it completely disregards normal points as $\mathcal{L}_{dir}$ is not applicable for them. Therefore, $\mathcal{L}_{dist}$ is indispensable in our offset prediction-based framework.

**Generating pseudo anomalies guided by *normal vectors* helps the normal representation learning:** A substantial performance drop is observed in "Variant 3", since moving points in random directions may produce unsuitable pseudo anomalies that confuse the model, resulting in less efficient learning. This indicates that the proposed Norm-AS is crucial for facilitating the extraction of normal representations. Besides, the detection performance of "Variant 3" further demonstrates the superiority of our offset prediction framework compared to reconstruction-based R3D-AD (77.2%).

## 4.6 ANALYSIS ON PATCH NUMBER

Fig. 4 reports the object-level and point-level AUC-ROC results *vs.* different patch numbers, which are average on fifteen categories ending in 0 of the Anomaly-ShapeNet dataset. The size of pseudo-abnormal regions is inversely correlated with the patch number $J$. An appropriate size is crucial for learning normal representations. Difficulty in predicting point offset for a region that is too large may hinder the model's convergence. Conversely, learning point offsets for a region that is too small may prevent the model from capturing sufficient normal representations. However, despite these effects, our method is generally less sensitive to the size of pseudo-abnormal regions. According to the presented results, the detection and localization performance reach their best when the patch numbers are 32 and 64, respectively. We set the patch number to 64 in our implementation to achieve the best detection performance, at the cost of a slight sacrifice in localization performance.

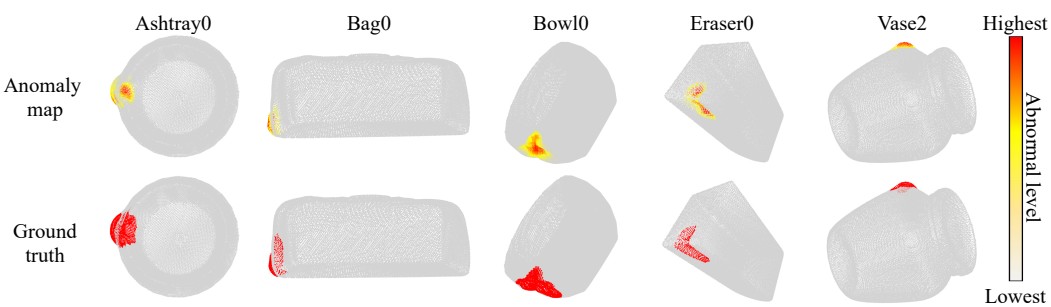

Figure 6: Qualitative results of localization on five categories of the Anomaly-ShapeNet dataset, where brighter color refers to a higher abnormal level.

### 4.7 ROBUSTNESS TO NOISY DATA

In real-world scenarios, the complexity of environments and the instability of equipment may result in scanned point clouds containing noise, i.e., noisy data. To analyze the robustness of our method with respect to noisy data, we conduct experiments on test samples containing Gaussian noise with a standard deviation of 0, 0.001, 0.003, and 0.005 (0 denotes clean data).

Selecting bottle0, 1, and 3 as illustrative categories, analysis results are presented in Fig. 5. It is observed that performance only drops slightly as the noise standard deviation increases. Additionally, the worst case of our method is still higher than competing methods tested on clean data (such as 73.3%, 73.7%, and 78.1% object-level AUC-ROC of R3D-AD on bottle0, 1, and 3). Such empirical results evidence the robustness of our method to noisy data. We visualize noisy point clouds in Fig. 8 of Appendix C.

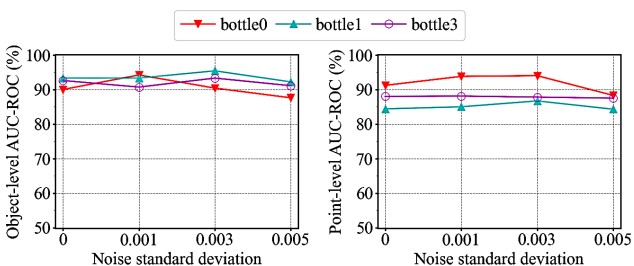

Figure 5: Detection and localization performance *vs.* noise with various standard deviations.

### 4.8 QUALITATIVE RESULTS

Fig. 6 illustrates anomaly maps for localization on five categories of the Anomaly-ShapeNet dataset. The anomaly map is obtained by performing the point-level scoring function $\phi(p_i)$. Evidently, our method precisely locates the abnormal regions, and also assigns relatively much lower abnormal levels to normal points. This validates the effectiveness of our method.

## 5 CONCLUSION

In this paper, we design a novel framework PO3AD based on point offset prediction to capture effective normal representations for 3D point cloud anomaly detection. Moreover, we propose an anomaly simulation method named Norm-AS guided by *normal vectors*, creating credible pseudo anomalies from normal samples to facilitate the distillation of normal representations. Extensive experiments conducted on the Anomaly-ShapeNet and Real3D-AD datasets evidence that our method outperforms the existing best methods.

**Limitations and future work.** It is imperative to note that our current design is still under the one-model-per-category learning paradigm, i.e., each category needs a specifically trained detection model, leading to prohibitive computational and storage. In future work, we intend to investigate the inter-category common patterns to explore a one-model-all-category learning paradigm for point cloud anomaly detection.

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

## A  VISUALIZATIONS OF OUR PSEUDO ANOMALIES

Fig. 7 presents visualizations of normal, real anomaly, and our pseudo anomaly samples. It is observed that our Norm-AS enables the creation of credible pseudo anomalies, which look very similar to real anomalies.

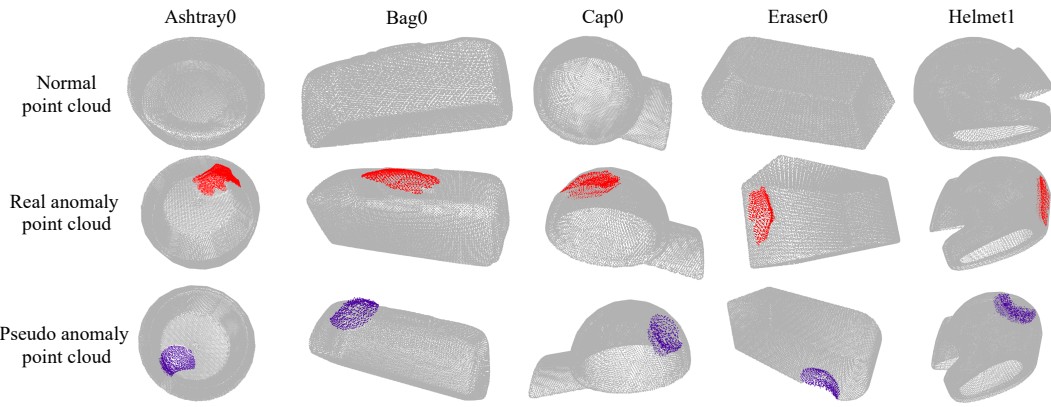

Figure 7: Visualizations of normal, real anomaly, and our pseudo anomaly samples.

## B  ADDITIONAL EXPERIMENTAL RESULTS

We report comparison object-level AUC-PR results on the Anomaly-ShapeNet dataset in Table 5. Evidently, our method achieves the best mean rank and significantly outperforms the second-best method by an average of 26.0% AUC-PR. Such experimental results evidence the superiority of our method.

Table 5: Comparison of object-level AUC-PR results on the Anomaly-ShapeNet dataset.

| Category | BTF (Raw) (CVPR 23') | BTF (FPFH) | M3DM (CVPR 23') | PatchCore (FPFH) (CVPR 22') | PatchCore (PointMAE) | CPMF (PR 24') | Reg3D-AD (NeurIPS 23') | IMRNet (CVPR 24') | Ours |
|---|---|---|---|---|---|---|---|---|---|
| ashtray0 | 57.8 | 65.1 | 63.2 | 44.5 | 67.9 | 45.3 | 58.8 | 61.2 | **99.9** |
| bag0 | 45.8 | 55.1 | 64.2 | 60.8 | 60.1 | 65.5 | 60.8 | 66.5 | **80.9** |
| bottle0 | 46.6 | 64.4 | 76.3 | 61.5 | 54.5 | 58.8 | 63.2 | 55.8 | **92.7** |
| bottle1 | 57.3 | 62.5 | 67.4 | 67.7 | 64.5 | 59.2 | 69.5 | 70.2 | **95.9** |
| bottle3 | 54.3 | 60.2 | 45.1 | 57.9 | 65.1 | 50.5 | 47.4 | 64.8 | **96.2** |
| bowl0 | 58.8 | 57.6 | 52.5 | 54.8 | 56.2 | 77.5 | 49.4 | 48.1 | **94.6** |
| bowl1 | 46.4 | 64.8 | 51.5 | 54.5 | 61.1 | 62.1 | 51.5 | 50.4 | **90.5** |
| bowl2 | 57.6 | 51.5 | 63.0 | 61.1 | 45.6 | 60.1 | 49.5 | 68.1 | **88.8** |
| bowl3 | 65.4 | 49.9 | 63.5 | 62.0 | 55.6 | 41.8 | 44.1 | 61.4 | **92.7** |
| bowl4 | 60.1 | 63.2 | 57.1 | 57.5 | 60.1 | 68.3 | 62.4 | 63.0 | **98.5** |
| bowl5 | 61.5 | 69.9 | 60.1 | 54.1 | 58.5 | 68.5 | 55.5 | 65.2 | **90.4** |
| bucket0 | 65.2 | 48.3 | 60.9 | 60.4 | 54.1 | 66.2 | 63.2 | 57.8 | **92.3** |
| bucket1 | 62.0 | 64.8 | 50.7 | 56.5 | 64.2 | 50.1 | 71.4 | 73.2 | **88.2** |
| cap0 | 65.9 | 61.8 | 56.4 | 58.5 | 56.1 | 60.1 | 69.3 | 71.1 | **84.1** |
| cap3 | 61.2 | 57.9 | 65.2 | 45.7 | 58.3 | 54.1 | 71.1 | 70.2 | **90.6** |
| cap4 | 51.5 | 54.5 | 47.7 | 65.5 | 72.1 | 64.5 | 62.3 | 65.8 | **87.6** |
| cap5 | 65.3 | 59.3 | 64.2 | 72.5 | 54.2 | 69.7 | 77.0 | 50.2 | **80.1** |
| cup0 | 60.1 | 58.5 | 57.0 | 60.4 | 64.2 | 64.7 | 53.1 | 45.5 | **87.9** |
| cup1 | 70.1 | 65.1 | 75.2 | 58.6 | 71.0 | 60.9 | 63.8 | 62.7 | **87.0** |
| eraser0 | 42.5 | 71.9 | 62.5 | 58.4 | 80.1 | 54.4 | 42.4 | 59.9 | **99.5** |
| headset0 | 37.9 | 53.1 | 63.2 | 70.1 | 51.5 | 60.2 | 53.8 | 70.1 | **76.5** |
| headset1 | 51.5 | 52.3 | 62.3 | 60.1 | 42.3 | 61.9 | 61.7 | 65.6 | **91.4** |
| helmet0 | 55.9 | 56.8 | 52.8 | 52.5 | 63.3 | 33.3 | 60.0 | 69.7 | **86.4** |
| helmet1 | 38.8 | 72.1 | 62.7 | 63.0 | 57.1 | 50.1 | 38.1 | 61.5 | **96.1** |
| helmet2 | 61.5 | 58.8 | 63.6 | 47.5 | 49.6 | 47.7 | 61.8 | 60.2 | **93.4** |
| helmet3 | 52.6 | 56.4 | 45.8 | 49.4 | 61.1 | 64.5 | 46.8 | 57.5 | **84.9** |
| jar0 | 42.8 | 47.9 | 55.5 | 49.9 | 46.3 | 61.8 | 60.1 | 76.0 | **91.5** |
| micro. | 61.3 | 66.2 | 46.4 | 33.2 | 65.2 | 65.5 | 61.4 | 55.2 | **80.3** |
| shelf0 | 62.4 | 61.1 | 66.5 | 50.4 | 54.3 | 68.1 | 67.5 | 62.5 | 68.0 |
| tap0 | 53.5 | 61.0 | 72.2 | 71.2 | 71.2 | 63.9 | 67.6 | 40.1 | **85.6** |
| tap1 | 59.4 | 57.5 | 63.8 | 68.4 | 54.2 | 69.7 | 59.9 | **79.6** | 70.9 |
| vase0 | 56.2 | 64.1 | **78.8** | 64.5 | 54.8 | 63.2 | 61.5 | 57.3 | 75.3 |
| vase1 | 44.1 | 65.5 | 65.2 | 62.3 | 57.2 | 64.5 | 46.8 | 72.5 | **78.9** |
| vase2 | 41.3 | 56.9 | 61.5 | 80.1 | 71.1 | 63.2 | 64.1 | 65.5 | **96.3** |
| vase3 | 71.7 | 65.2 | 55.1 | 48.1 | 45.5 | 58.8 | 65.1 | 70.8 | **90.2** |
| vase4 | 42.8 | 58.7 | 52.6 | 77.7 | 58.6 | 65.5 | 50.5 | 52.8 | **82.4** |
| vase5 | 61.5 | 47.2 | 63.3 | 51.5 | 58.5 | 51.8 | 58.8 | 65.4 | **87.9** |
| vase7 | 54.7 | 59.2 | 64.8 | 62.1 | 65.2 | 43.2 | 45.5 | 60.1 | **97.1** |
| vase8 | 41.6 | 62.4 | 46.3 | 51.5 | 65.5 | 67.3 | 62.9 | 63.9 | **83.3** |
| vase9 | 48.2 | 63.8 | 65.1 | 66.0 | 63.4 | 61.8 | 57.4 | 46.2 | **90.4** |
| Average | 54.9 | 59.8 | 60.3 | 58.8 | 59.5 | 59.7 | 58.4 | 62.1 | **88.1** |
| Mean rank | 6.5 | 5.3 | 5.1 | 5.6 | 5.6 | 5.1 | 5.6 | 4.6 | **1.0** |

# C  VISUALIZATIONS OF NOISY DATA

We illustrate the visualizations of a clean point cloud and its noisy variants with various standard deviations in Fig 8. It is observed that as the noise standard deviation grows, the point cloud surface becomes progressively less smooth.

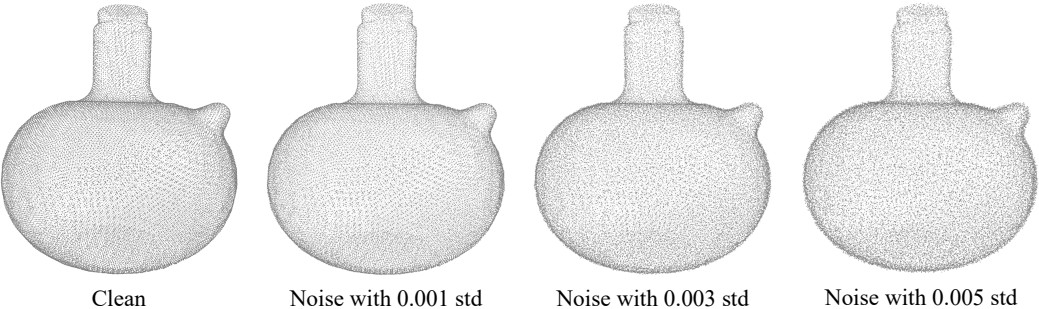

Clean   Noise with 0.001 std   Noise with 0.003 std   Noise with 0.005 std

Figure 8: Visualizations of clean, and noisy point clouds with various standard deviations (std).