# OpenReview forum: "PO3AD: Predicting Point Offsets toward Better 3D Point Cloud Anomaly Detection"
_ICLR.cc/2025/Conference — ICLR 2025 Conference Withdrawn Submission_

### Official Review · Reviewer_1cRn · 2024-10-17

**Soundness:** 2
**Presentation:** 2
**Contribution:** 3
**Rating:** 5
**Confidence:** 4

**Summary:**

In this study, an anomaly detection method PO3AD based on point cloud reconstruction is proposed, which attempts to reduce the attention of the model on the less important points by giving more attention to the anomalies, and concentrate the model performance on the abnormal part. The contribution of the model was demonstrated by the Real3D-AD dataset and the Anomaly-Shapenet dataset.

**Strengths:**

This study presents a valuable framework that has the potential to serve as a benchmark for research in this area. The advantages are as follows:

1. The model has strong scalability.

2. The model idea is simple but work.

 3. The starting point of the model is reasonable.

**Weaknesses:**

If there are no mistakes in my understanding, I am more inclined to reject this paper, which is due to the following reasons:

 1. The paper tries to prove that the network allocates more performance to focus on handling exceptions, but it does not seem sufficient to show that the model achieves this by just (c) and (d) of Figure 1.

 2. The model is trained by the pseudo-exception generation method, which we observe is a mode used by most reconstruction methods. The paper does not compare the advantages and disadvantages of the proposed NORM-AS with other methods.

3. This paper directly uses MinkUNet as an anomaly recovery network. How is this backbone superior in anomaly detection tasks compared with other models? Why is it not modified to better adapt to 3DAD tasks?

4.The writing of the paper has some miscellaneous.

**Questions:**

The main argument:
The point of view of this paper does not seem to be well proved that their argument is consistent with the model, although I think that their framework allocates more attention to the generated exception part, but I think it is still to be studied. In addition, the paper does not seem to prove well that their exception generation and directly invoked MinkUNet exception recovery framework is superior, and further experiments may be needed.

Compared with the existing methods, the method proposed in this paper lacks too many convincing details. I have the following questions:

The main problem:
1. Why only 15 types of ablation were selected for the experiment? You know, it seems unreasonable to selectively extract 15 classes from a large dataset of 40 classes. Maybe I need a complete ablation experiment of 40 classes. A complete multi-class ablation experiment helps us better understand the various parts of the model.

2. Have you considered comparing your NORM-AS with other methods for 3DAD tasks? For example, the exception generation method in IMRNet and R3D in your related work.

3. Have you considered comparing more point prediction frameworks to show that your MinkUNet choice is correct? Your 3.1.1 provides multiple backbone methods, please explain why you chose MinkUNet. You can do this through qualitative experiments or, more simply, persuasive verbal explanations.

4. Would you consider giving more arguments for your idea of giving outliers more attention, such as visual visualizations (similar to heat maps of outliers)?

5. If the horizontal time span of Figure 1 (d) is the same, then the curve only seems to indicate that your model has difficulty converging to the anomaly.

Secondary issues:
1. Does the design of the loss function have better weights to get better results?

2. Reconstruction methods tend to perform poorly in terms of training time, have you considered comparing the time performance considerations of your method with other reconstruction methods?

Other questions (questions that do not affect ratings) :

1. Is your visualization of Figure 6 a qualitative analysis drawn from the scores generated during the reasoning process, or is it just expressing abnormal ideas?

2. Do you consider expanding the model into a complete framework? I think your model has considerable potential, the backbone of which can be replaced by most point prediction models. If you can do this, this paper will be better.

Ignore these and the results seem unconvincing. The idea of the paper seems to be sound, but we still need more experiments to prove that "the model does pay more attention to the outliers". Perhaps readers need a more explanatory quantitative indicator to judge how to pay more attention to the outliers. Readers may still need experiments with predictive frameworks other than MinkUNet to understand this idea.

Some improvements that do not affect ratings:

1. You may want to consider using a clearer abstract. Your abstract is a bit complex, and it may be difficult for some readers who lack the basics to understand the whole process.

2. Your description of performance can be made clearer by separating P-AUROC and I-AUROC.

The most important advice: I don't want to be stingy with my praise, but your paper has the potential to serve as a more robust framework, so I may have to give a weak reject to help you make this paper better. Experiment more with your offset prediction method and your exception generation method, this may be the next more powerful method. This paper needs more experiments, and is capable of multiple feature extraction methods like PatchCore (of course, it is different prediction network in your paper, you can abstract a more general and robust prediction model from a variety of existing point cloud migration prediction models). I believe this paper is a good one.

---

### Official Review · Reviewer_vMEo · 2024-10-24

**Soundness:** 3
**Presentation:** 3
**Contribution:** 3
**Rating:** 5
**Confidence:** 5

**Summary:**

This paper proposes to predict point offsets for point cloud anomaly detection. To facilitate the training process, a novel point cloud anomaly generation strategy is proposed. The authors also designed offset loss for training. Experimental results on Real3D and Anomaly-ShapeNet effectively demonstrate the superior anomaly detection performance of the proposed method.

**Strengths:**

The proposed method achieves significantly better point cloud anomaly detection performance in comparison to other alternatives.

This paper proposes a new scheme for point cloud anomaly detection, i.e., predicting the point offset.

**Weaknesses:**

I fail to fully grasp the motivation for predicting the offset of points instead of reconstructing all points. Through the discussion between Line 80-97, it seems that a scheduler, or focal losses, may contribute to adaptively focusing on different kinds of points. Shall you provide more explanations for your proposed point-offset prediction scheme?

Are these generated anomalies realistic enough? do you have any idea how to mitigate the overfitting risk?

The experiments are conducted on Real3D and AnomalyShapeNet. Why don’t you also conduct experiments on the more sophisticated dataset MVTec 3D?

Regarding the implementation details, the author utilizes MinkUNet34C for feature extraction, which differs from previous methods, like M3DM which utilizes PointMAE. Can you explain the reason for this design and conduct ablation on the backbone? Also, here you don't utilize the downsampling operation, why?

Why are there no point-level results for Real3D-AD?

I will consider improving my scores if you can address these questions.

**Questions:**

See the weakness.

---

### Official Review · Reviewer_EqGH · 2024-10-30

**Soundness:** 2
**Presentation:** 2
**Contribution:** 1
**Rating:** 3
**Confidence:** 5

**Summary:**

The paper highlights that current 3D Anomaly Detection (3D-AD) research primarily focuses on designing unsupervised reconstruction tasks, which often overlook abnormal regions in point clouds. To address this, the paper introduces offset prediction into the 3D-AD task. By predicting the offsets of pseudo-abnormal points, the model enhances its representation of normal points. Additionally, the paper proposes a point cloud pseudo-anomaly synthesis method, making the training process more efficient. Experimental results demonstrate that PO3AD is currently a SOTA method.

**Strengths:**

+ The paper is the first to introduce offset prediction into the 3D-AD task and effectively demonstrates its benefits.
+ Although the code lacks environment specifications and a README file, providing detailed code for reference is beneficial.
+ Experimental results on Anomaly-ShapeNet and Real3D-AD are compelling and persuasive.

**Weaknesses:**

+ The paper resembles an empirical investigation. While offset prediction is effective for the 3D-AD task and the loss decreases similarly to reconstruction methods, there is a lack of in-depth analysis. Maybe there should be some theoretical justifications of why offset prediction is more effective than reconstruction, or ablation studies on the offset prediction approach.
+ Despite the importance of offset prediction in the paper, there is no introduction to related work in this area.  The authors could include a brief overview of offset prediction techniques in traditional point cloud domain, and discuss why apply such technique to 3D-AD.
+ The design for the anomaly detection task is limited, and Norm-AS is similar to R3D-AD, offering little novelty.

**Questions:**

+ Currently, it seems that reconstruction-based methods are not popular in 3D-AD, with much of the research focusing on KNN combined with memory banks. If the model has better representation capabilities, could it also be applied in a KNN-plus-memory-bank framework?
+ Why does $L_{dir}$ significantly influence the experiments in the ablation study? How does it perform on Real3D-AD?

---

### Official Review · Reviewer_VUsJ · 2024-11-05

**Soundness:** 3
**Presentation:** 2
**Contribution:** 2
**Rating:** 5
**Confidence:** 3

**Summary:**

This paper proposes a novel method to address the problem of 3D point cloud anomaly detection. The
 method focuses on learning point offsets to better capture 3D normal data features and accurately identify
 anomalies. Furthermore, the paper introduces a point cloud augmentation technique based on normal vectors,
 which can effectively enhance the training efficiency. Experimental results on two datasets demonstrate that the
 proposed method performs better than the current state-of-the-art methods. Thus, the main contribution of this
 paper is to propose a new method to solve the problem of 3D point cloud anomaly detection and prove its
 effectiveness and superiority

**Strengths:**

The paper presents a novel approach to address the problem of outlier detection in 3D point clouds, demonstrating a certain degree of innovation and practicality. The research method and experimental design in this paper are rigorous, and the dataset used is comprehensive. At the same time, the results of the paper effectively demonstrate the effectiveness and superiority of the proposed method.

The language expression of the paper is clear, with a logical structure that is reasonable, easy to understand, and readable. Additionally, the paper provides detailed experimental results and analysis, enabling readers to better understand the specific implementation and effect of the method.

The point cloud anomaly detection method proposed in this paper holds significant value and importance for practical applications, especially in scenarios where anomaly detection of 3D point cloud data is required. This method can provide an effective solution.

**Weaknesses:**

The method was only tested on two datasets, so more datasets are needed to verify its generalization power.

The method relies on the simulation of normal data to generate pseudo-anomaly data, which may cause some pseudo-anomalous data to be less realistic, thereby affecting the performance of the model.

The method only considers the direction and distance of point deviation, without taking into account other important factors, such as the positional relationships between points.

In summary, while this method has certain advantages, it also has some limitations that need to be further improved and refined.

**Questions:**

I hope the authors can further explore the application scope and limitations of this method, as well as conduct a comparative analysis with other methods. Additionally, the authors could consider adding more experimental data and adjusting parameters to further improve the effectiveness of the method.

---

### Note · Authors · 2024-11-13

I have read and agree with the venue's withdrawal policy on behalf of myself and my co-authors.